# Objective Comparison of Achievement Motivation and Competitiveness among Semi-Professional Male and Female Football Players

**Ricardo de la Vega** [1,*], **Jorge Gómez** [2], **Raquel Vaquero-Cristobal** [3], **Javier Horcajo** [4] **and Lucía Abenza-Cano** [3]

1. School of Education, Autonomus University of Madrid, 28049 Madrid, Spain
2. Health Economy Motricity and Education (HEME), Faculty of Sport Science, University of Extremadura, 10003 Caceres, Spain; jorge.gomez@managuafc.ni
3. Faculty of Sport Science, Catholic University San Antonio of Murcia, 30007 Murcia, Spain; rvaquero@ucam.edu (R.V.-C.); labenza@ucam.edu (L.A.-C.)
4. Department of Social Psychology and Methodology, Autonomous University of Madrid, 28049 Madrid, Spain; javier.horcajo@uam.es
* Correspondence: ricardo.delavega@uam.es

**Abstract:** The aim of this study was to examine differences in achievement motivation (measured with the Objective Achievement Motivation Test, OLMT, Schuhfried®) and competitiveness between male and female semi-professional football players. The OLMT objectively assessed three constructs regarding achievement motivation: motivation through personal goals, aspiration level, and motivation through competition. In addition, competitiveness was measured with the self-reported Competitiveness-10 Questionnaire. Finally, participants' performance was assessed by three expert observers in each of ten matches. Thirty-eight football players (men = 27; women = 11) participated in the present study, and no significant differences were found in the Levene test when comparing men and women with respect to the scores obtained in the different measures used in our research. Significant differences were found in the motivation through competition ($p = 0.021$) as well as in self-reported competitiveness ($p = 0.020$) as a function of gender, with males showing higher values in both cases. No gender differences were found in aspiration level ($p = 0.283$) or motivation through personal goals ($p = 0.897$). Moreover, age and player performance did not modulate gender differences on any measures. No significant correlation was found between motivational measures and performance. In conclusion, it should be noted that the only variable on which gender differences emerged was the level of competitiveness, such that males scored higher than females on both objective and self-reported measures.

**Keywords:** OLMT; achievement motivation; competitiveness; computerized assessment; football

## 1. Introduction

There is a broad scientific consensus in considering the importance of psychological variables, along with technical, tactical, and physiological, on sports performance [1,2], which are considered a very important structure for the development of the player [3,4]. Of all the psychological aspects that could play a role in performance, research has shown that motivation can counteract the decrease in performance caused by fatigue [5] and has an important influence, along with psychophysiological, cognitive, and emotional components, on an athlete's maximum possible effort [5,6]. Specifically, achievement motivation is based on the demonstration of high competence and/or avoiding the demonstration of low ability [7], and establishing a positive relationship between achievement motivation and sport performance [2,8].

Task-oriented motivation is necessary to succeed on a task, regardless of whether or not a reward is present [9–11]. In this case, success depends on the effort exerted and

is linked to the improvement of one's task-relevant skills [9,12,13], with mistakes being a part of the improvement process [13]. According to numerous studies, a task-oriented motivational climate favors the performance of football teams [3,14].

In addition, with ego-oriented motivation, individuals seek to achieve success by focusing their attention on the pride experienced in outperforming others [6]. In fact, this motivational driver can produce greater satisfaction as the athlete assumes a perspective focused on competition and overcoming rivals, instead of focusing on overcoming themselves [9,14]. Taken together, high-level teams seem to be characterized by a greater ego orientation, leading to a greater likelihood of developing performance-oriented group climates. For their part, lower-level teams would be more task-oriented, which would favor improvement-oriented team climates with more cooperative than competitive approaches [6,15]. In this sense, there is also evidence on the relationship between ego-task orientations and athletes' perception of success [9,12,13,15]. A high perception of success increases the level of effort and persistence in the task [16], which leads to greater motivation and enjoyment of success [17].

For the interests of our research, the study of achievement motivation from a Cattell's personality-centered perspective is assumed [18]. In motivation research, a fine distinction is sometimes made between motive and motivation. According to Schneider and Schmalt [19], achievement motivation is always a state and the achievement motive a disposition. This differentiation is relevant because most of the studies carried out to date do not establish a clear difference between the two concepts [20]. This implies that a punctual evaluation of achievement motivation is used, and the evaluation of the dispositional tendency, related to personality, of achievement motivation is left aside. It is precisely this dispositional and objective assessment that allows the OLMT to be incorporated into Cattell's personality assessment tradition. In this perspective, personality trait is measured by behavioral indicators and not by subjective test [19,21].

Regarding competitiveness, this research has focused on assuming the trait approach framed in the Murray, Atkinson, and McClelland achievement theory [22–24]. These authors consider that in achievement environments, such as sports, people act driven by stable personality factors and by situational factors. Dispositional factors or motives are the motive for achieving success and the motive for avoiding failure. According to these authors, these factors remain stable over time, are universal (because in achievement environments, actions obey the need for achievement) and are independent of each other because a person can have a high need for achievement. Avoid failure, but not to achieve success [25].

As this relates to the present research, evidence suggests that gender differences exist between male and female athletes in terms of motivational factors, specifically achievement motivation [11,26]. For example, studies have shown that, within the context of sports, men have higher ego-oriented motivation [13] and higher motivation through competition than women [11,27], whereas women have a higher aspiration level and are better at setting goals than men [10]. There are no gender differences in motivation to avoid failure [27].

In psychological research, specifically regarding achievement motivation, the vast majority of studies have used subjective self-report instruments [28], despite being a measurement that has been criticized for its limitations [29,30], as it can be easily faked [31] and runs the risk of generating unreliable results [32]. Despite these drawbacks, self-report measures also have some advantages, such as their low economic cost and high ease of use [33–35]. Notably, only a few studies have used objective instruments to measure athletes' motivation [11], despite the fact that objective measures can be more reliable than self-reports in some specific circumstances [30]. Importantly, some authors advocate the joint use of both (self-reported and objective) measures, in order to obtain a broader assessment of athletes' motivation [10,11,32].

Much of the empirical evidence on achievement motivation in football has relied on self-report measures [36,37] and focused on samples made up of male football players [38], who compete in youth or training categories [39,40], and on amateur adults [41]. Very few

studies have been conducted using a female population [42,43], on which computerized psychological measures have been used to collect data via the new technologies that have recently emerged [44]. The present research aimed to address this gap regarding research on achievement motivation in the context of semi-professional football players.

In sum, studies that analyze achievement motivation in football players, using objective instruments, and exploring differences as a function of gender, seem relevant and necessary. Therefore, the main objective is to analyze gender differences in objectively measured achievement motivation and self-reported competitiveness between male and female semi-professional soccer players, comparing the relationship between self-reported and objectively measured competitiveness.

## 2. Materials and Methods

### 2.1. Design

A descriptive cross-sectional design was performed in accordance with STROBE guidelines. The institutional ethics committee reviewed and authorized the protocol designed for data collection in accordance with the World Medical Association Code (code number: CEI-106-160). The guidelines of the Declaration of Helsinki were followed throughout the process.

After requesting permission from the club managers, the coaches of the teams participating in the study were informed of the general aims of this research, the procedure to be followed, and the implications of this study. All participants were also informed of the aim of the study and the procedure to be followed, assuring them of the absolute confidentiality of the data and results obtained. Once informed consent had been provided by all participants involved in this study, the researchers then provided information regarding the place, day, and time of the tests.

The calculations for establishing the sample size were performed using Rstudio 3.15.0 software. The significance level was set at $\alpha = 0.05$. The standard deviation (SD) was established based on previous studies for competitiveness (mean SD = 0.55) [27]. With an estimated error (d) of 0.17 for a sample n = 38; 0.21 for a sample n = 27 and 0.32 for a sample n = 11.

### 2.2. Participants

The sample consisted of 38 semi-professional football players (male = 27; female = 11). The mean age of the participants was $25.5 \pm 4.90$ years ($M_{male} = 28.4 \pm 4.60$; $M_{female} = 22.6 \pm 5.20$) and belonged to the same football club. Male and female players from the same club have been selected to control for the possible effect of contextual and cultural variables. In the case of the men's team, the number of years playing football was $16.3 \pm 2.80$ years and at the time the research was carried out, the team was in its third season with the same technical staff. In the case of the woman's team, the number of years playing football was $10.5 \pm 4.40$ years and at the time the research was carried out, the team was in its third season with the same technical staff. Sample selection was incidental for convenience and followed the relevance criteria. Inclusion criteria were: (1) the main sport of all the players had to be football, (2) a minimum playing time of ten years, and (3) a current season's registration in the third division of the Spanish football federation (semi-professional level).

### 2.3. Procedure

Prior to the start of each assessment, participants received specific instructions on how to perform each test, including familiarization trials with the Objective Achievement Motivation Test (OLMT, Vienna Test System VTS, Schuhfried®). The OLMT system itself is prepared to propose a series of tests that ensure the correct understanding of the test. In case of detecting an increase in the response latency before the appearance of stimuli, or random responses outside the expected threshold values, the system blocks and does not allow progress. Once this phase was completed, they were given the Competitiveness Questionnaire-10 [27], which was completed individually. Subsequently, the OLMT test

was administered. Finally, three expert observers evaluated the performance of the players during each of the 10 soccer matches based on a standardized observation sheet.

### 2.4. Instrumentation

To assess competitiveness, Remor's Competitive Questionnaire-10 [27] was administered. This is a self-report instrument with ten questions about motivation associated with sport competitiveness, and included two factors: motivation for success (MS) ($\alpha$ = 0.66) and motivation to avoid failure (MAF) ($\alpha$ = 0.66), and a final competitiveness score (MS–MAF).

The OLMT was used to collect objective data on achievement motivation. The VTS is a standardized [45], computerized, valid, and reliable [46] test battery that has been used for psychometric assessment [45] and to gain insight into the cognitive level and neurophysiology of human movement [47,48], developed by Schuhfried GmbH (Mödling, Austria) [11,32]. The OLMT seeks to assess the amount of individual achievement motivation [11] based on four constructs [11,31] that are measured in three subtests that require performing the same exercise on a computer screen. The respondent moves along a prescribed route cell by cell by pressing two buttons repeatedly. He/She has ten seconds in which to advance as far as possible. After the first phase (motivation through the activity -task orientation-), the respondent is asked to set targets and then to achieve them (motivation through goals -task orientation-). Finally, the respondent is pitted against a virtual opponent whose speed is slightly above the speed achieved by the respondent in the first two subtests (motivation through competition -ego orientation-). The respondent is now asked to outdo the opponent [45]. The OLMT was conducted individually, without the presence of others, as this can have detrimental effects on focused attention [49].

Finally, a record sheet was used in which three expert observers assessed the players' performance during ten football matches. The observers were football coaches studying to obtain the highest qualification (UEFA Pro level), without prior knowledge of the players who were part of the study. Performance was subjectively assessed using a single item scale from 0 to 10, with 0 corresponding to low performance and 10 to high performance. Each of those observed was asked to assess the performance they observed in the players evaluated in each game, offering a final assessment of their perception of performance. To do this, they were asked to take into account perceived aspects of the technical, tactical, physical, and psychological performance of the players. A low sports performance would be equivalent to a technical, tactical, physical, and psychological performance perceived as low, while the opposite would happen with high sports performance.

After calculateing each player's average score across all ten matches for each individual observer, total scores for each observer were then averaged among the three observers in order to obtain more reliable data. Pearson's correlation (r = 0.78) between the scores of the three evaluators was significant (*p* = 0.009).

### 2.5. Data Analysis

First normality analyses were performed using the Shapiro–Wilk test. The results indicated that the measures were normal, according to which a parametric statistical test were applied (*p* > 0.05). The Student's *t*-test was used to analyze whether there were differences between gender on each measure The effect size was calculated using Cohen's d coefficient. A value lower than 0.2 was considered a low effect size; a value between 0.2 and 0.4 was considered a low-moderate effect; a value between 0.4 and 0.6 was considered a moderate effect; a value between 0.6 and 0.8 was considered moderate-high effect; and a value higher than 0.8 was considered a high effect (Cohen, 1988). In addition, ANCOVA was performed to analyze the influence of age and sport performance as covariates on the differences between genders on the motivational variables. Given the obtained parametric distribution for age and sport performance, we conducted a Pearson bivariate correlation analysis to examine the relationship between participants' OLMT scores (dependent variables) on three constructs (i.e., motivation through personal goals, aspiration level, and motivation through competiton), as well as the scores on motivation

for success and motivation to avoid failure from the Competitiveness-10 Questionnaire. The statistical package SPSS 21.0 was used for all analyses. In a complementary way, a generalizability analysis was carried out to assume that the estimated results were reliable and generalizable by the SAGT v1.0 software [50].

## 3. Results

Descriptive analyses of the study variables and differences between gender are shown in Table 1. Significant differences were found in motivation through competition ($p = 0.021$) and self-reported competitiveness ($p = 0.020$) as a function of gender, with males showing higher scores in both cases. No gender differences were found in aspiration level ($p = 0.283$) and motivation through personal goals ($p = 0.897$). Cronbach's alpha coefficients were analyzed for the total sample: motivation for success (MS) ($\alpha = 0.72$) and motivation to avoid failure (MAF) ($\alpha = 0.70$), both values were acceptable.

**Table 1.** Comparison in motivation through competition, aspiration level, motivation through personal goals, and self-reported competitiveness between males and females.

|  | Gender | Mean ± SD | Dif Mean | t | p | 95%ICC | d Cohen |
|---|---|---|---|---|---|---|---|
| Motivation through competition | Male | 5.05 ± 0.94 |  |  |  |  |  |
|  | Female | 3.31 ± 2.30 | 2.92 ± 2.69 | 2.41 | 0.021 | −2.55; 8.40 | 0.99 |
|  | Total | 4.54 ± 2.14 |  |  |  |  |  |
| Aspiration level | Male | 3.05 ± 2.09 |  |  |  |  |  |
|  | Female | 2.22 ± 2.15 | 0.82 ± 0.75 | 1.091 | 0.283 | −0.71; 2.35 | 0.39 |
|  | Total | 2.81 ± 2.11 |  |  |  |  |  |
| Motivation through personal goals | Male | 0.06 ± 3.54 |  |  |  |  |  |
|  | Female | −0.11 ± 4.18 | 0.17 ± 1.33 | 0.131 | 0.897 | −2.53; 2.88 | 0.04 |
|  | Total | 0.01 ± 3.68 |  |  |  |  |  |
| Self-reported competitiveness | Male | 1.21 ± 0.46 |  |  |  |  |  |
|  | Female | 0.76 ± 0.62 | 0.45 ± 0.18 | 2.44 | 0.020 | 0.07; 0.82 | 0.82 |
|  | Total | 1.08 ± 0.54 |  |  |  |  |  |

When analyzed as covariates, players' age and sport performance did not significantly modulate gender differences in motivation through competition (F = 1.493, $p = 0.232$; and F = 0.368, $p = 0.549$, respectively), aspiration level (F = 0.023, $p = 0.881$; and F = 0.689; $p = 0.414$, respectively), motivation through personal goals (F = 1.771, $p = 0.194$; and F = 0.011, $p = 0.919$, respectively), or self-reported competitiveness (F = 1.792, $p = 0.191$; and F = 0.053, $p = 0.819$, respectively).

Regarding objectively measured motivation (i.e., motivation through competition) and subjectively self-reported competitive motivation, no significant correlation was obtained within the total sample (r = 0.205; $p = 0.217$), nor among males (r = 0.083; $p = 0.680$) or females (r = 0.080; $p = 0.815$).

Finally, no significant correlation was found between the player's sport performance and any of the motivational variables analyzed either in the total sample or for either gender (Table 2).

**Table 2.** Correlations between sport performance and motivation through competition, aspiration level, motivation through personal goals, and self-reported competitiveness.

|  |  | Motivation through Competition | Aspiration Level | Motivation through Personal Goals | Self-Reported Competitiveness |
|---|---|---|---|---|---|
| Sport performance | Male | r = −0.010; p = 0.963 | r = −0.175; p = 0.436 | r = −0.061; p = 0.788 | r = 0.074; p = 0.743 |
|  | Female | r = −0.345; p = 0.328 | r = −0.121; p = 0.738 | r = −0.172; p = 0.635 | r = −0.038; p = 0.916 |
|  | Total | r = 0.096; p = 0.600 | r = −0.065; p = 0.722 | r = −0.049; p = 0.790 | r = 0.138; p = 0.453 |

## 4. Discussion

One of the most relevant findings of the present research was the discovery of significant differences in the OLMT' assessment of motivation through competition among semi-professional football players as a function of gender, with males scoring higher than females. Relevantly, significant differences were also found in Remor's [27] self-reported Competitiveness-10 Questionnaire. Prior research has found similar results in other studies using different populations and activities [26,51,52]. Thus, Ong [11] argues that the higher scores in competitiveness of males compared to those of females may be due to the presence of a competitor increasing a dominance instinct of males, subsequently motivating them to try to achieve higher performance. Similar results were found by Hepler and Witte [53], who reported that male athletes focus more on the outcome than the process than female athletes, perhaps influenced by cultural aspects. In this sense, it is relevant to reflect on the results found in the context of Spanish football, where there may be a tradition that has emphasized the importance of males' football compared to females'. Due to the changes produced in these cultural aspects, it would be interesting to replicate this study not only with a larger sample, but also in a few years to see if changes occur.

Furthermore, traditional scientific literature has suggested that the sporting arena provides an ideal setting for male athletes to achieve success and to demonstrate their superiority over opponents [54]. As a complement to these explanations, it seems relevant to us to point out the influence of cultural aspects on the levels of aspiration and competitiveness of male and female soccer players. It is possible that these results change if players from the highest categories are analyzed. If the component that influences the most is environmental/cultural, the results in competitiveness could be similar [54]. If, on the contrary, competitiveness is a variable more dependent on personality, then we could continue to find differences between both genders. Regarding the first perspective, which puts the emphasis on environmental aspects, the perspective of analysis of motivational climates stands out. In this perspective, motivational climates are perceived differently by men and women, and this is reflected in self-report measures [26,55]. These authors recommended coaches to accentuate task-oriented climates, although it is known that ego-oriented climates can provide positive effects for both genders.

Regarding aspiration level, the results of the present research showed that in a team sport such as football, no significant differences emerged between men and women.

Moreover, the results regarding the absence of statistically significant differences in the motivation through personal goals between men and women are consistent with those found by Tomczak [13], using the Polish version of the Perception of Success Questionnaire (POSQ). However, these results contrast with those obtained by Miller et al. [56], where men obtained higher scores both in competitiveness (ego-oriented motivation) and in motivation through personal goals (task-oriented motivation) using the Perceived Motivational Climate in Sport Questionnaire (PMCSQ).

Regarding the limitations of the present research, the sample size (especially for females) was somewhat small, so future studies with larger sample sizes are necessary to replicate the results of the present study. In any case, the considerably lower percentage of semi-professional females' football teams in Spain must be taken into account, which limited the possibility of expanding the sample. It should also be taken into account that the research team chose to select a club that had a semi-professional-level males and females teams, which greatly limited the possibility of accessing a larger sample. This decision was justified in order to ensure very equivalent exposure in both teams to the most similar possible environmental conditions. As noted above, it would be worth investigating whether the gender differences in aspiration level and the little or no difference in the other constructs are replicated at other competitive levels. This would allow us to analyze the role of the environment regarding achievement motivation as a personality variable, with important effects on the selection and development of sports talents. Another limitation of this study is the possibility of comparing the results for two fundamental reasons: firstly,

the few studies that have used the OLMT for the evaluation of achievement motivation and, secondly, the absence of studies that have used this methodology in football.

In addition, no significant differences in motivation through personal goals were observed between genders. This contrasts with the results of Pulido-Pedrero et al. [10] in elite combat athletes, in which women performed better in setting personal goals than men using the OLMT as an objective assessment measure. The differences between the two studies could be due to the differences between the sports analyzed, and future research is needed to investigate the influence of the sport modality practiced on the differences between genders in motivational processes. Finally, it is important to emphasize the future realization of longitudinal designs that overcome the limitations of the cross-sectional design used, as well as the possibility of using non-incidental sampling that allows increasing the representativeness of the sample used.

With respect to the generalizability design, it is orthogonal with three facets (gender, age, and observed performance) and partially nested (two nested facets, gender and age). Three used facet designs were [AGE][GEND]/[PERF], [AGE]/[GEND][PERF], and [PERF][AGE]/[GEND]. The absolute G values found were the following: [AGE][GEND]/[PERF] = 0.406; [PERF][AGE]/[GEND] = 0.156; [GEND][PERF]/[AGE] = 0.277. The G indices in the three estimated models were adequate, accounting for the stability of both measures. An explained variance percentage of 18.54% was found associated with the categories nested in the criteria ([AGE]:[GEND]), being 11.33% for ([GEND]:[PERF]) and 8.4 % for ([AGE]:[PERF]).

## 5. Conclusions

In conclusion, it should be noted that there was a significant difference between genders in the level of competitiveness, being higher in male football players on both the objective (i.e., motivation through competition) and self-reported competitiveness measures, with no significant differences being observed in the aspiration level, and motivation through personal goals. It is desirable to expand knowledge in team sports and specifically in football, investigating the causes of these gender differences and their implications. It is also important to reflect on the practical implications of the results obtained to emphasize the importance that coaches should give to achievement motivation evaluated from the perspective of personality psychology.

**Author Contributions:** Conceptualization, R.d.l.V., J.G. and J.H.; methodology, R.d.l.V., R.V.-C. and L.A.-C.; formal analysis, R.d.l.V. and R.V.-C.; investigation, J.G.; writing—original draft preparation, R.d.l.V., R.V.-C., L.A.-C. and J.H.; writing—review and editing, R.d.l.V. and J.H.; project administration, J.H. and R.d.l.V.; funding acquisition, J.H. and R.d.l.V. All authors have read and agreed to the published version of the manuscript.

**Funding:** This research was supported by the Ministerio de Ciencia e Innovación (Spain) [Grant number: PID2020-116651GB-C33/AEI/10.13039/501100011033].

**Institutional Review Board Statement:** The study was conducted in accordance with the Declaration of Helsinki, and approved by the Institutional Ethics Committee) of Autonomus University of Madrid (protocol code CEI-106-160).

**Informed Consent Statement:** Informed consent was obtained from all subjects involved in the study.

**Data Availability Statement:** The datasets are available from the corresponding author on reasonable request.

**Conflicts of Interest:** The authors declare no conflict of interest. The funders had no role in the design of the study; in the collection, analyses, or interpretation of data; in the writing of the manuscript, or in the decision to publish the results.

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
