# Peer review of "Objective Comparison of Achievement Motivation and Competitiveness among Semi-Professional Male and Female Football Players"

_sustainability, doi:10.3390/su14095258_

Round 1
Reviewer 1 Report
Dear author, first of all thank you for your submission. I believe I did a good job, that the article explores an emerging theme related to motivation in women's football that.
Although the article is well structured, I believe that some improvements at the beginning of the introduction should be made.
Below are my specific comments.
page 1 - line 8 - I believe that the author wants to mention that "performance in football is not exclusively explained by physical, technical and tactical abilities since psychological factors also assume a preponderant role in performance".
The idea is the same but I believe it can be clarified. not necessarily in the way I suggest, I just think it should be clearer to read.
In addition, increase the number of references to the subject at the end of the sentence to clearly emphasize this idea
Line 10 - 12 - Dear author, maybe it would be important to mention here the importance of psychophysiological indicators to better explain the potential relationship between motivation and fatigue?
Author Response
We sincerely appreciate the comments made by the reviewer. We have made a significant effort to respond to your comments and we hope that it meets your expectations. The changes made to the manuscript as a result of their comments are set out below:
1. page 1 - line 8 - I believe that the author wants to mention that "performance in football is not exclusively explained by physical, technical and tactical abilities since psychological factors also assume a preponderant role in performance".
Answer: We have made the suggested changes. (Page 1 Line 8-9)
2. Line 10 - 12 - Dear author, maybe it would be important to mention here the importance of psychophysiological indicators to better explain the potential relationship between motivation and fatigue?
Answer: We have made the suggested changes. (Page 1 Line 13-14)
Reviewer 2 Report
Dear authors.
In the introductory part, expressions such as "some studies", "studies have shown" are often used and reference is made to the same source (example source 11, which is a single one).
L59-64: The purpose is confusing. It needs to be clarified. There are 3 directions and the study claims that it has a single purpose. The description made indicates 3 directions of the study and makes the purpose unclear. Adapt and be clearer.
A single purpose cannot be expressed by several actions "to analyze", "to determine", "to relate" and "to examine".
L86-87 is repeated "and belonged to the same football"
L94 - it is necessary to describe what OLMT represents here, at L94 and deleted from L97, because here appears the first reference
L95 Competitiveness-10 Questionnaire [19], is it the same as Remor's [19] Competitive-10 Questionnaire, from L101? Because they are not described in the same way in the text
L105 The Objective Achievement Motivation Test, Schuhfried® (OLMT) has already been described as L97, should not be repeated here… use OLMT
The same goes for the Vienna Test System, the abbreviation VTS must be changed to L94, not L106
The procedure needs to be described more clearly, both with the variables evaluated and as a way of analysis.
The subjective evaluation of the 3 experts by the scale from 0 to 10 should be more clearly delimited: what did they notice as low performance or high performance? it does not appear from your explanations
In the Discussions section, they should be oriented and argue what you have measured as variables. Because there are many discussions that do not affect the variables you are evaluating in this article. Comparisons on similar studies or on studies with similar variables should have been made in articles published also on football, not on other sports, in order to be relevant.
The number of female subjects is irrelevant to generalize an idea, to draw a conclusion, and to support comparative analysis.
Author Response
Dear Reviewer,
We sincerely thank you for your review work. We have reviewed all your comments and have substantially improved our article. We hope that the important effort we have made will be valued positively by you.
Below we present the changes made and the answers to the observations that you have made to us.
1. In the introductory part, expressions such as "some studies", "studies have shown" are often used and reference is made to the same source (example source 11, which is a single one).
Answer: Indeed, the reference to Ong's research (11) is very important to us. The justification lies in the importance of the evaluation of achievement motivation from the perspective of the study of computerized objective evaluation (Cattell´s perspective), being the research team of this author one of the ones that has worked the most in this topic. In this sense, we are interested in emphasizing this study due to the possibility of comparing its results with ours, thus trying to overcome the most important limitations of evaluation through questionnaires.
2. L59-64: The purpose is confusing. It needs to be clarified. There are 3 directions and the study claims that it has a single purpose. The description made indicates 3 directions of the study and makes the purpose unclear. Adapt and be clearer.
A single purpose cannot be expressed by several actions "to analyze", "to determine", "to relate" and "to examine".
Answer: The wording of the objectives has been modified, simplifying their reading and understanding in the direction indicated by the reviewer.
3. L86-87 is repeated "and belonged to the same football".
Answer: One of the phrases that was indeed repeated has been removed.
4. L94 - it is necessary to describe what OLMT represents here, at L94 and deleted from L97, because here appears the first reference
Answer: The wording of the paragraph has been profoundly modified to facilitate its understanding, following the instructions of the reviewer. (L111-117)
5. L95 Competitiveness-10 Questionnaire [19], is it the same as Remor's [19] Competitive-10 Questionnaire, from L101? Because they are not described in the same way in the text.
Answer: It is actually the same test. Its name has been unified.
6. L105 The Objective Achievement Motivation Test, Schuhfried® (OLMT) has already been described as L97, should not be repeated here… use OLMT.
Answer: We appreciate the observation. It has been corrected in the sense pointed out by the reviewer and we have used only OLMT.
7. The same goes for the Vienna Test System, the abbreviation VTS must be changed to L94, not L106
Answer: We appreciate the observation. It has been corrected in the sense pointed out by the reviewer.
8. The procedure needs to be described more clearly, both with the variables evaluated and as a way of analysis.
Answer: (L111-121) We have made a significant effort to facilitate the understanding of the procedure. It has been completely modified to respond to the observation made and allow its replicability.
9. The subjective evaluation of the 3 experts by the scale from 0 to 10 should be more clearly delimited: what did they notice as low performance or high performance? it does not appear from your explanations.
Answer: (L148-153). A more detailed explanation has been added, giving complete information to reviewers 2 and 3.
10. In the Discussions section, they should be oriented and argue what you have measured as variables. Because there are many discussions that do not affect the variables you are evaluating in this article. Comparisons on similar studies or on studies with similar variables should have been made in articles published also on football, not on other sports, in order to be relevant.
Answer: (L304-307). An attempt has been made to provide a response to this observation with the inclusion of a new explanatory paragraph.
11.
Answer: We acknowledge this limitation of the study, as reflected in lines 298-301. (L300-306) We have added relevant information to justify the limitation of the sample size of the female team.
Reviewer 3 Report
The manuscript tackles an interesting topic in the field of sport psychology. However, there are serious concerns about theoretical and methodological points that require be meaningfully addressed.
Comment 1. Abstract is missed. Please, provide a meaningful abstract indicating background, objectives, method, main results and conclusions.
Comment 2. Introduction: There is evidence making us think this research falls into the Achievement Goal Theory (Ames, 1995; Nicholls, 1984); however, the authors did not indicate the conceptualization under this framework. Indeed, the dichotomy for goal orientation represents the classical paradigm distinguishing task-orientation goals from performance-orientation goals. This paradigm has been moved toward a 2 x 3 achievement goal framework by combining valence (approach or avoidance) with definition (task, self or others) (Elliot et al., 2011). On other hand,
Comment 3: Lines 22-32: There is a general sense of confusion when talking about goal orientation and motivational climate (Lines 22-32). It is recommended to be more precise when using the different variables comprising Achievement Goal Theory.
Comment 4. There is already evidence about gender differences in achievement motivation with professional athletes, even using observational measures. This is, indeed, indicated by the authors (line 33-39). It is, therefore, needed to provide a stronger rationale for the study of these gender differences in the specific sport of football. Is football so different from another sport?
Comment 5. The study contribution requires to be meaningful strengthened.
Comment 6. Line 52: Please, add a full stop before V.: adults [31] Very few
Comment 7. The competitiveness is introduced for the first time in objectives.
Comment 8. Participants: The number of male and female footballer is unbalanced, as well as the belonging to the same club could largely jeopardize the obtained results. This point must be seriously tackled. Moreover, more detailed information is needed to obtain a better characterization of the participating sample such as sports background, years working with their current coach, or number of training hour a week. Additionally, it is needed to specify the type of sampling method used to recruit the participating footballers.
Comment 9. Instruments: I have serious doubt about the suitability of the instruments measuring competitiveness and achievement motivation. On the one hand, the measure of competitiveness is centered on motivation toward competitiveness, which truly failed to assess the competitiveness. Regardless of these points, it is needed to include more information such as the stem that introduces the questionnaire, item to exemplify each dimension, as well as the Likert-type scale. Furthermore, it should be noted that Cronbach’s alpha scores equal or higher than 0.80 must be obtained when examining mean differences among groups (Viladrich et al., 2017). On the other hand, although the objective achievement motivation test could represent a measure of achievement motivation, this failed to measure the most current paradigm of achievement motivation theory. Regarding agreement among experts, it is recommended to estimate Fleiss’ Kappa’s coefficient instead of Pearson’s correlation (Fleiss et al., 2003). In particular, the Fleiss’ kappa coefficient is a specific measure of agreement among three or more coders (Fleiss et al., 2003). Moreover, it is needed to detail the specific training and previous experience of the observers involved in this process.
Comment 10. Data analysis: It is surprising that data followed a normal distribution, even the ones from the Competitive-10 Questionnaire collected by a 3-point Likert scale. Please, estimate Cronbach’s alpha for every dimension comprising competitiveness. Regarding ANCOVA, it is required to provide a strong rationale for considering age ad sport performance as covariables. Additionally, eta partial squared must be added as effect-size measure for this type of test.
Comment 11. Discussion. It is needed to discuss the results in accordance with the Spanish context of football.
Comment 12. Limitations: Additional limitations should also be mentioned. For instance: cross-sectional design or type of used sampling method.
Comment 13. Implications for coaching practice are missed.
Comment 14. Comment: References: Please, revise the format (Palatino linotype, 10) for all references.
References
Ames, C. (1995). Achievement goals, motivational climate, and motivational processes. In Motivation in sport and exercise (pp. 161–176). Human Kinetics Books.
Elliot, A. J., Murayama, K., & Pekrun, R. (2011). A 3 × 2 achievement goal model. Journal of Educational Psychology, 103(3), 632–648. https://doi.org/10.1037/a0023952
Fleiss, J. L., Levin, B., & Paik, M. C. (2003). Statistical methods for rates and proportions (3rd ed.). Wiley.
Nicholls, J. G. (1984). Achievement motivation: Conceptions of ability, subjective experience, task choice, and performance. Psychological Review, 91(3), 328–346. https://doi.org/10.1037/0033-295X.91.3.328
Viladrich, C., Angulo-Brunet, A., & Doval, E. (2017). A journey around alpha and omega to estimate internal consistency reliability. Anales de Psicologia/ Annals of Psychology, 33(3), 755–782. https://doi.org/10.6018/analesps.33.3.268401
Author Response
We appreciate your thorough review of the manuscript. We have tried to answer each of the questions you ask us. The changes made are presented below and we hope that they will be positively valued by the reviewer.
1. Abstract is missed. Please, provide a meaningful abstract indicating background, objectives, method, main results and conclusions.
Answer: We have added the summary in this version. In the previous one it was in the section enabled on the magazine's own platform.
2. Comment 2. Introduction: There is evidence making us think this research falls into the Achievement Goal Theory (Ames, 1995; Nicholls, 1984); however, the authors did not indicate the conceptualization under this framework. Indeed, the dichotomy for goal orientation represents the classical paradigm distinguishing task-orientation goals from performance-orientation goals. This paradigm has been moved toward a 2 x 3 achievement goal framework by combining valence (approach or avoidance) with definition (task, self or others) (Elliot et al., 2011). On other hand,
Answer: (L35-45). As the reviewer points out, we have specified the theoretical background that supports our research. For us it´s very important to start from a dispositional perspective of personality, since it is one of the fundamental novelties of this research. Traditional models have focused on the study of motivational manifestations towards achievement from a state approach, while our approach focuses on a trait approach similar to the one proposed by Cattell in 1975. In addition to adding an explanatory paragraph, added relevant references from leading authors in this perspective.
3. Comment 3: Lines 22-32: There is a general sense of confusion when talking about goal orientation and motivational climate (Lines 22-32). It is recommended to be more precise when using the different variables comprising Achievement Goal Theory.
Answer: (L23-34)The revision carried out is very positively valued. It has been decided to clarify the fundamental aspects that, indeed, were quite confusing. In any case, it is a paragraph that does not focus on the perspective assumed at work, but it is important insofar as it allows us to focus on the relationship between the motivational tendency and the team climates that are generated. In this sense, being semi-professional teams, competitiveness and achievement motivation are expected to be relevant components of the players who are part of the research.
4. Comment 4. There is already evidence about gender differences in achievement motivation with professional athletes, even using observational measures. This is, indeed, indicated by the authors (line 33-39). It is, therefore, needed to provide a stronger rationale for the study of these gender differences in the specific sport of football. Is football so different from another sport?.
Answer: (L35-45) The novel contribution of our study lies in the use of a theoretical background focused on the personality trait approach from a Cattell´s perspective. (see L). The prototype for all subsequent objective personality tests similars to the ones we used is the Objective Test Battery (OA-TB 75), a selection of tests based on Cattell published by Häcker, Schmidt and Schwenkmezger (1975).
5. Comment 5. The study contribution requires to be meaningful strengthened.
Answer: We have responded to this comment in the previous sections.
6. Comment 6. Line 52: Please, add a full stop before V.: adults [31] Very few
Answer: We have made the suggested change.
7. Comment 7. The competitiveness is introduced for the first time in objectives.
Answer: (L74-82). A paragraph has been added to frame the importance of competitiveness. This paragraph allows to understand the selection of the selected test (Competitiveness-10).
8. Comment 8. Participants: The number of male and female footballer is unbalanced, as well as the belonging to the same club could largely jeopardize the obtained results. This point must be seriously tackled. Moreover, more detailed information is needed to obtain a better characterization of the participating sample such as sports background, years working with their current coach, or number of training hour a week. Additionally, it is needed to specify the type of sampling method used to recruit the participating footballers.
Answer: We appreciate the comment made. The information provided on the different aspects pointed out by the reviewer has been completed and added to manuscript.
9. Comment 9. Instruments: I have serious doubt about the suitability of the instruments measuring competitiveness and achievement motivation. On the one hand, the measure of competitiveness is centered on motivation toward competitiveness, which truly failed to assess the competitiveness. Regardless of these points, it is needed to include more information such as the stem that introduces the questionnaire, item to exemplify each dimension, as well as the Likert-type scale. Furthermore, it should be noted that Cronbach’s alpha scores equal or higher than 0.80 must be obtained when examining mean differences among groups (Viladrich et al., 2017). On the other hand, although the objective achievement motivation test could represent a measure of achievement motivation, this failed to measure the most current paradigm of achievement motivation theory. Regarding agreement among experts, it is recommended to estimate Fleiss’ Kappa’s coefficient instead of Pearson’s correlation (Fleiss et al., 2003). In particular, the Fleiss’ kappa coefficient is a specific measure of agreement among three or more coders (Fleiss et al., 2003). Moreover, it is needed to detail the specific training and previous experience of the observers involved in this process.
Answer: We believe that the complete revision of the document has brought clarity to the justification of the instruments used. They have been selected by congruence with the starting theoretical background, trying to compare the results obtained by self-report (Competitiveness-10), with respect to those obtained by computerized objective evaluation (OLMT). Respect to Kappa coefficient, indeed Fleiss' arbitrary guidelines (0.75 is excellent) seem to be cited most often but Kappa is intrinsically nonlinear, does not account for error well, and retains an influence of bias, so kappa has been reviewed that would not be preferable to correlation as a standard independent measure of agreement. We understand the reviewer's observation but we request to maintain the analysis carried out.
10. Comment 10. Data analysis: It is surprising that data followed a normal distribution, even the ones from the Competitive-10 Questionnaire collected by a 3-point Likert scale. Please, estimate Cronbach’s alpha for every dimension comprising competitiveness. Regarding ANCOVA, it is required to provide a strong rationale for considering age ad sport performance as covariables. Additionally, eta partial squared must be added as effect-size measure for this type of test.
Answer: (L255-257). Cronbach's alpha coefficients have been analyzed for the total sample in each of the dimensions of the Competitiveness-10. ANCOVA was carried out because it is a statistical procedure that makes it possible to eliminate the heterogeneity caused in the variable of interest (achievement motivation scores) by the influence of one or more quantitative variables (age and competitive level). In our case, age can generate a source of heterogeneity due to the differences between the male and female teams (mean values and standard deviation have been added). The same happens with the control of the possible heterogeneity generated by the competitive level of the two teams. Respect to effect size, we consider that Cohen´s d is an appropiate effect size for the comparison between two means. Conceptually, the d family effect sizes are based on the difference between observations, divided by the standard deviation of these observations.
11. Comment 11. Discussion. It is needed to discuss the results in accordance with the Spanish context of football.
Answer: (L342-346) A paragraph has been added to discuss the results in accordance with the Spanish context of football.
12. Comment 12. Limitations: Additional limitations should also be mentioned. For instance: cross-sectional design or type of used sampling method.
Answer: (L392-395). A paragraph has been added about the additional limiations noted by the reviewer.

Round 2
Reviewer 2 Report
Thanks to the authors for their efforts, for the corrections made to the recommendations.
The idea is to be appreciated, but the way of presentation is a bit difficult. The presentation of data requires a lot of attention, but not impossible. Probably due to the psychological aspect of the article.
But, in the end, the study offered by you can bring value in the field, if the readers will have the patience and attention to go through and determine the essence of the research.
Success!
Reviewer 3 Report
Thank you so much for responding to every comment proposed. I feel satisfied with the labour completed.